# Oral Health-Related Quality of Life in People with Achalasia

**DOI:** 10.3390/medicina56060286

**Published:** 2020-06-11

**Authors:** Marcel Hanisch, Sabrina Wiemann, Lauren Bohner, Susanne Jung, Johannes Kleinheinz, Sebastian Igelbrink

**Affiliations:** Department of Cranio-Maxillofacial Surgery, Research Unit Rare Diseases with Orofacial, Manifestations (RDOM), University Hospital Münster, Albert-Schweitzer-Campus 1, Building W 30, D-48149 Münster, Germany; s_wiem07@uni-muenster.de (S.W.); lauren.bohner@ukmuenster.de (L.B.); susanne.jung@ukmuenster.de (S.J.); johannes.kleinheinz@ukmuenster.de (J.K.); sebastian.igelbrink@ukmuenster.de (S.I.)

**Keywords:** rare diseases, oral health-related quality of life, OHRQoL, OHIP-14, achalasia

## Abstract

*Background and Objective*: The oral health-related quality of life (OHRQoL) of patients with achalasia has not been evaluated to date. Therefore, the aim of this study was to assess the OHRQoL of patients with achalasia and to get information about the time taken for diagnosis and oral symptoms. *Materials and Methods:* The study was conceived of as an anonymous epidemiological survey study in people with achalasia in order to assess their OHRQoL in each case. For this, a questionnaire was developed consisting of free-text questions and of the standardized German version of the OHIP-14 questionnaire. *Results:* In total, forty-four questionnaires were analyzed including 31 female and 13 male participants. Regardless of gender, the mean age was 50.57 years (range: 17–78). Of the surveyed individuals, seventy-nine-point-five-five percent had been diagnosed between 25 and 60 years of age. The period from the first signs of the disease to diagnosis was 6.15 years, irrespective of gender. The overall OHIP-14 score without gender differentiation was 8.72 points (range 0–48); the mean score of female participants was 11.13 (range: 0–48), and the OHIP score of male participants was 3.15 on average. Two participants reported oral symptoms. *Conclusions:* The already known problem of the delayed diagnosis of rare diseases was also confirmed in the case of achalasia. Females with achalasia seemed to be significantly affected by lower OHRQoL than males with achalasia and women of the general population. Demineralization of the tooth structure was described in two participants.

## 1. Introduction

Achalasia is a rare disease characterized by degeneration of nerve fibers involving the Auerbach’s plexus of the esophagus [1,2,3,4]. As a consequence, non-peristaltic contractions and the subsequent inability of esophageal musculature to relax after swallowing are the main features of the disease. Clinically, dysphagia, regurgitation, weight loss, and malnutrition may occur [4,5]. In 27–42% of the cases, reflux after consumed solid foods and liquids is reported [6].

Achalasia can be classified as primary or secondary. Primary achalasia is diagnosed based on clinical symptoms and typical radiological findings, after ruling out secondary achalasia. In turn, secondary achalasia is caused by organic esophageal stenosis such as tumorous infiltrations or extra esophageal diseases [7]. Although the disease etiology is still unknown [3,7], autoimmune and inflammatory diseases, such as Sjogren’s syndrome, rheumatoid arthritis, or Marfan syndrome, may be related to achalasia [8]. However, as the disease is rare [9], it is often misunderstood and interpreted as simple reflux [5]. Subsequently, patients often only receive appropriate treatment when they have had the disease for a long time [7].

Since orofacial anatomical structures are directly related to esophageal structures, it is not uncommon that patients with achalasia present oral manifestations. Besides the dysphagia of solids and liquids after swallowing [8], a long-term manifestation is dental erosion, which is characterized as the loss of hard dental tissue due to the pH of the regurgitated acid. According to the severity of the dental manifestations, it results in aesthetic and functional complications, which require surgical and prosthetic treatments [10,11].

Previous studies showed that rare diseases presenting oral manifestations may contribute to the decrease of oral health-related quality of life [12,13]. Thus, in order to conduct diagnosis and treatment properly, clinicians should be aware of clinical manifestations of achalasia. However, the literature regarding clinical features of patients with achalasia is still scarce. Based on this, the present study purposes to assess clinical features and oral manifestations of people diagnosed with achalasia. Additionally, oral health-related quality of life was assessed using the Oral Health Impact Profile 14 (OHIP-14). The questionnaire measured the frequency of 14 different functional and psychosocial influences on oral health-related quality of life [14]. Differences regarding the disease presentation and diagnosis between genders were statistically assessed. The null hypothesis was that there was no difference in clinical manifestations, diagnosis, and OHIP-score between male and female patients.

## 2. Materials and Methods

### 2.1. Study Design

The study was approved by the ethics committee of the Medical Association of Westphalia-Lippe and the Westphalian Wilhelms University in Münster (Data of approval: 2016; Ref. No. 2016-006-f-S) and designed as an anonymous epidemiological survey of people with achalasia.

In order to evaluate their respective oral health-related quality of life, a questionnaire consisting of free-text questions about age, gender, date of diagnosis, time taken for diagnosis, and any oral symptoms related to the underlying disease “achalasia” was developed.

In addition, the standardized German version of the OHIP-14 (Oral Health Impact Profile) questionnaire was used to assess the oral health-related quality of life [14]. The answers to the questions, which were related to the participants perception over the past month, were assigned according to standardized numerical values: 0 = never, 1 = rarely, 2 = occasionally, 3 = often, and 4 = very frequently. In this regard, an overall score of 0 to a maximum of 56 points could be obtained, in such a way that the higher the numerical value, the worse the oral health-related quality of life.

The study was conceived of as a cohort study in people affected with achalasia. In February 2016, the questionnaire was circulated in digital format to achalasia self-help groups (Achalasie Selbsthilfe e.V.) listed under the umbrella organization of chronic rare disease self-help groups in the Federal Republic of Germany ACHSE e.V. (Allianz Chronisch Seltener Erkrankungen e.V. (Alliance of Chronic Rare Diseases)) and displayed during the annual meeting of the self-help group in April 2016. Achalasie Selbsthilfe e.V reported 450 members. Returns were accepted up until February 2018. This questionnaire was acquired for a previous study performed by our research group, but a new analysis was performed.

### 2.2. Participants

People affected by achalasia in the Federal Republic of Germany were eligible to participate in this study. The requirements consisted of age above 16 years and membership in the self-help group “Achalasie Selbsthilfe e.V.”.

### 2.3. Data Source

In addition to age, sex, and disease, possible oral symptoms, age at the time of diagnosis, and the period from the first appearance of the disease until diagnosis were recorded, and the individual OHIP values were calculated.

### 2.4. Statistical Analysis

Statistical analysis was performed using the software SPSS 22.0 (IBM Group; Armonk, NY, USA). Participants were divided into two groups according to gender: male and female. The mean differences of the following variables were compared between groups: age during diagnosis, time between diagnosis, and first symptoms, as well as OHIP values. First, the Kolmogorov–Smirnov test was used to test the normality of data. Age during diagnosis showed adherence to normality, whereas “time between diagnosis and first symptom” and OHIP values did not show adherence to normality. Levene’s test was used to evaluate the homogeneity of variance, which was significant for all groups. The mean difference between gender was compared using Student´s t-test for independent samples and the Mann–Whitney test with a statistically significant difference at *p* = 0.05.

## 3. Results

### 3.1. Participants

A total of 44 questionnaires were evaluated, from 31 female (70.45%) and 13 male (29.55) participants. The gender-independent age was 50.57 years (range: 17–78); the average age of female participants was 50.71 years (range: 17–78), and the male participants were on average 50.23 years old (range: 35–69). The descriptive data and statistical analysis are described in Table 1.

### 3.2. Age at Diagnosis and Time Until Diagnosis

The gender-independent average age at which “achalasia” was diagnosed was 41.00 years (range: 15–68 years). Female participants were on average 43.35 years old (range: 15–68 years), and the men were on average 35.38 years old (range: 24–49 years).

The age of diagnosis was distributed across all participants, with 35 participants in the time window between 25 and 60 (79.55%) years, four participants younger than 25 years, and five participants over 60 years as their age at the time of diagnosis. The earliest diagnosis was a 15-year-old, and the latest was a sufferer diagnosed at the age of 68.

The mean period of time between the first symptoms of the disease and diagnosis, regardless of gender, was 6.15 years (range: 0.25–44 years). Among the female participants, the mean age was 6.94 years (range: 0.42–44 years), while for men, it was 4.25 years (range: 0.25–13 years). There was no statistically significant difference between gender for the age during diagnosis (*p* = 0.068) and for the time between first symptom and diagnosis (*p* = 0.502).

### 3.3. Oral Symptoms

For the question about oral symptoms related to their disease, forty-two of the participants provided information, but only two participants reported oral symptoms (4.76%). Both participants described demineralization of the tooth structure. The other 40 participants did not report oral symptoms associated with achalasia (95.24%).

### 3.4. OHIP Values

The mean OHIP-14 total score regardless of gender amounted to 8.72 points (range: 0–48). For the female participants, the mean score was 11.13 points (range: 0–48), and for male participants, an OHIP mean score of 3.15 was obtained (range: 0–17), showing a statistical difference (*p* = 0.004) between them.

## 4. Discussion

On average, it takes seven years before a rare disease is correctly diagnosed [15]. According to Orphanet [6], most sufferers are diagnosed when they are aged between 25 and 60. In this study, around 80% of participants described diagnosis in this period and thus confirmed the well-known figures, with participants younger than 16 years old excluded from the study beforehand. Because of the rarity of achalasia, this rare disease is also often only diagnosed at a late stage [7]. These findings were shown in this study, since the majority of participants were above 25 years at the time of diagnosis. Furthermore, a period of 6.15 years could be ascertained between the first symptoms of the disease and the “achalasia” diagnosis, with men apparently diagnosed earlier (4.25 years) than women (6.94 years). Thus, the previously-known problem of the delayed diagnosis of rare diseases [13] was also confirmed for achalasia, and an additional psychological burden for those affected and their families as a result of possible misdiagnosis could be assumed [16].

The aim of this study was to find out whether any oral symptoms could be determined in people affected by achalasia, such as dental erosion caused by fermented foods (lactic acid) as described by Moatzzez et al. [17]. An obstructive esophagus causes food fermentation, and in turn, regurgitation of fermented food causes dental erosion [17]. Conversely, only two participants (4.76%) out of all those surveyed here reported dental erosion of the tooth structure. However, none of the participants in the study had been clinically examined by the authors, so the actual rate of lactic acid-related damage to the tooth structure may be higher than that indicated by the study participants. To obtain meaningful results, future specific clinical investigations should be conducted.

John et al. [18] described a gender-independent average of 4.09 for the OHIP-14 overall score in the overall German population. Men in the general population indicated a poorer oral health-related quality of life (4.50) than women (3.72) [18]. Standard values were used to interpret the levels of impaired oral health-related quality of life for individuals and groups of people in comparison to the degree of impaired oral health-related quality of life in the general population [18]. The OHIP-14 total score of 8.72 for people with achalasia calculated in this study therefore differed from the values for the general population. A separate analysis of the three dimensions of the OHIP was not recommended by the authors of the German OHIP-14 version [11]. People with achalasia consequently exhibit a poorer oral health-related quality of life than the general population. It is known that people with rare diseases are affected by a reduced OHRQoL. This has already been reported in other studies, which have analyzed the OHRQoL in people with rare diseases [13]. In accordance, a poor health-related quality of life was seen in patients with abnormalities affecting the oral cavity and its functionality, including musculoskeletal disorders and oral lesions with pain conditions, difficulty in mastication, bad hygiene, or hampered social well-being [19,20,21,22]. However, one reservation must be borne in mind: the respondent population (50.57 years) was generally older on average than the population surveyed when analyzing the oral health-related quality of life in the general population (43.34 years) [18].

Further examination of the results, however, clearly indicated that this applied only to the female study participants. These showed an OHIP-14 total score of 11.13 points and were therefore significantly higher than the values described by John et al. [18] for women in the general population (3.72). The data obtained here therefore indicated that women (11.13 points) with achalasia suffered from a poorer oral health-related quality of life than men (3.15 points) with achalasia.

### Limitations of the Study

The data were collected on the basis of self-assessment in a sample of people with achalasia. The participants were, on average, older than the population surveyed when analyzing the oral health-related quality of life in the general population. However, the study provided insight into oral symptoms and the oral health-related quality of life in people with achalasia, and therefore, future studies can be based on these data.

## 5. Conclusions

Based on the findings of this study, it was possible to conclude that: -People suffering from achalasia had a delayed diagnosis of the disease;-The incidence of oral manifestations described by the participants was low and limited to dental erosion;-Females with achalasia seemed to be significantly affected by lower OHRQoL than males with achalasia.

## Figures and Tables

**Table 1 medicina-56-00286-t001:** Descriptive statistics of the study population.

Gender	Age at Diagnosis	Time Until Diagnosis	OHIP Values
Male			
Mean (SD)	35.38 (10.16)	4.25 (4.00)	3.15 (4.94)
Median (interquartile range)	34 (24.50–45.00)	3.00 (1.50–5.00)	0.00 (0.00–6.00) *
Female			
*n*	30	30	30
Mean (SD)	43.35 (13.96)	6.94 (9.22)	11.13 (10.87)
Median (interquartile range)	42.50 (32.75–55.25)	4.5 (1.37–10.00)	8.50 (3.50–15.50) *
*p*-value	0.068	0.502	0.004

* Means statistically significant difference between male and female participants. SD = standard deviation.

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
