# Peer review of "Oral Health-Related Quality of Life in People with Achalasia"

_medicina, 2020, doi:10.3390/medicina56060286_

Round 1
Reviewer 1 Report
Few comments and questions:
A. More detail study design description:
- Prospective or retrospective, cohort or other type of study.
- More detail description about patient's selection into this study.
- More detail explanation, how achalasia was confirmed (clinical manifastation only, or instrumental investigations: X ray study, esophagomonometry, CT scan and others.
- When did patients answer the questionare: before treatment or after it.
Author Response
We would like to thank the Editor and the Reviewers for revising our manuscript [medicina-795190] entitled “Oral health related quality of life in people with achalasia“ and the constructive points discussed. The helpful comments and suggestions for improving the manuscript have been incorporated into the revised version and all changes were highlighted “Track Changes”. In this letter, we provide a point-by-point response to each addressed comment and hope the manuscript is now suitable for publication in the Medicina.
Reviewer 1:
Comments and Suggestions for Authors
Few comments and questions:
More detail study design description:
Prospective or retrospective, cohort or other type of study.
Answer: the study is a cohort study. We added this information on line 69, page 2.
More detail description about patient's selection into this study.
Answer: the participants had to be members of the self-help group “Achalasie Selbsthilfe e.V.” and be affected by achalasia. We added this information on lines 79-81, page 2.
More detail explanation, how achalasia was confirmed (clinical manifastation only, or instrumental investigations: X ray study, esophagomonometry, CT scan and others.
Answer: since we do not have a complete insight into the medical records of every participant, we cannot say whether the diagnosis was made solely on the basis of clinical symptoms or additionally with imaging procedures.
When did patients answer the questionare: before treatment or after it.
Answer: the participants are under medical treatment because of the underlying disease "achalasia". However, the data were collected independently of the therapy.
Reviewer 2:
Comments and Suggestions for Authors
On the other side I must underline that patients with achalasia do not suffer from acid reflux but from regurgitation of saliva and undigested food and I wonder if the non-acid reflux can damage thoot structure as the acid one. In the discussion for example, is cited Prader Willis syndrome in wich demineralisation is caused by acid gastro esophageal reflux due to an hypotonic cardias and gastric dilatation. So, please differentiate reflux from the classic regurgitation.
I appreciate the idea for the paper wich is interesting and original. Many patients with foregut disturbance as motility disorders or GE reflux (GER) can suffer from oral health disturbance and incidence is still unknown.
- Answer: you are absolutely right. Achalasia is related to dental erosion and the cause of the erosion is fermented food (lactid acid) and not regurgitated gastric juice. We added this information and included the reference [line 140-142], Refrence: Moazzez R, Anggiansah A, Botha AJ, Bartlett D.Moazzez R, et al. Association of achalasia and dental erosion. 2005 Nov;54(11):1665-6.
We added this information in line 137-144:….“The aim of this study was to find out whether any oral symptoms could be determined in people affected by achalasia, such as dental erosion caused fermented foods (lactid acid) as described by Moatzzez et al. [13]. An obstructive oesophagus causes food fermentation, and in turn regurgitation of fermented food causes dental erosion [13]. Conversely, only two participants (4.76%) out of all those surveyed here reported dental erosion to the tooth structure. However, none of the participants in the study had been clinically examined by the authors, so the actual rate of lactid acid related damage to the tooth structure may be higher than that indicated by the study participants. To obtain meaningful results, future specific clinical investigations should be conducted”…
The explanation of why a patient with rare disease should be prone to oral health damage is not clear and an explanation should be proposed in general and in particular with achalasia.
Answer: we revised this statement in the introduction section: “It is known that people with rare diseases are often affected by a reduced oral health-related quality of life (line 44-46) and explained the reason why people with achalasia are prone to oral health damage in the discussion section (line 137-144).
another point: line 34 the prevalence of achalasia is not 1/10 000 please correct the data with updated citations.
Answer: we corrected the prevalence and included two updated citations (line 34).
Citations:
- Samo S, Carlson DA, Gregory DL, Gawel SH, Pandolfino JE, Kahrilas PJ. Incidence and prevalence of achalasia in central Chicago, 2004-2014, since the widespread use of high-resolution manometry. Clin Gastroenterol Hepatol. 2017;15:366–373.
- Sadowski DC, Ackah F, Jiang B, Svenson LW. Achalasia: Incidence, Prevalence and Survival. A Population-Based Study. Neurogastroenterol Motil. 2010;22:256–261.
Reviewer 2 Report
I appreciate the idea for the paper wich is intresting and original. Many patients with foregut disturbance as motility disorders or GE reflux (GER) can suffer from oral health disturbance and incidence is still unknown.
On the other side I must underline that patients with achalasia do not suffer from acid reflux but from regurgitation of saliva and undigested food and I wonder if the non-acid reflux can damage thoot structure as the acid one. In the discussion for example, is cited Prader Willis syndrome in wich demineralisation is caused by acid gastro esophageal reflux due to an hypotonic cardias and gastric dilatation. So, please differentiate reflux from the classic regurgitation. The explanation of why a patient with rare disease should be prone to oral healt damage is not clear and an explanation should be proposed in general and in particular with achalasia.
another point: line 34 the prevalence of achalasia is not 1/10 000 please correct the data with updated citations.
Author Response

(The authors gave the same response as above.)
